# AdaCoder: Adaptive Prompt Compression for Programmatic Visual Question Answering

## ABSTRACT

Visual question answering aims to provide responses to natural language questions given visual input. Recently, visual programmatic models (VPMs), which generate executable programs to answer questions through large language models (LLMs), have attracted research interest. However, they often require long input prompts to provide the LLM with sufficient API usage details to generate relevant code. To address this limitation, we propose AdaCoder, an adaptive prompt compression framework for VPMs. AdaCoder operates in two phases: a compression phase and an inference phase. In the compression phase, given a preprompt that describes all API definitions in the Python language with example snippets of code, a set of compressed preprompts is generated, each depending on a specific question type. In the inference phase, given an input question, AdaCoder predicts the question type and chooses the appropriate corresponding compressed preprompt to generate code to answer the question. Notably, AdaCoder employs a single frozen LLM and pre-defined prompts, negating the necessity of additional training and maintaining adaptability across different powerful black-box LLMs such as GPT and Claude. In experiments, we apply AdaCoder to ViperGPT and demonstrate that it reduces token length by 71.1%, while maintaining or even improving the performance of visual question answering.

## CCS CONCEPTS

• **Computing methodologies → Multimedia**; **Computer vision**; **Natural language processing**.

## KEYWORDS

Visual programmatic models, Code generation, Visual question answering, Large language models, Prompt compression.

**ACM Reference Format:**
Anonymous Author(s). 2024. AdaCoder: Adaptive Prompt Compression for Programmatic Visual Question Answering. In *Proceedings of Proceedings of the 32th ACM International Conference on Multimedia (MM '24)*. ACM, New York, NY, USA, 9 pages. https://doi.org/XXXXXXX.XXXXXXX

## 1 INTRODUCTION

Visual question answering (VQA), which aims to automatically provide answers to questions related to visual content, is a challenging research topic in the fields of multimedia analysis, computer vision,

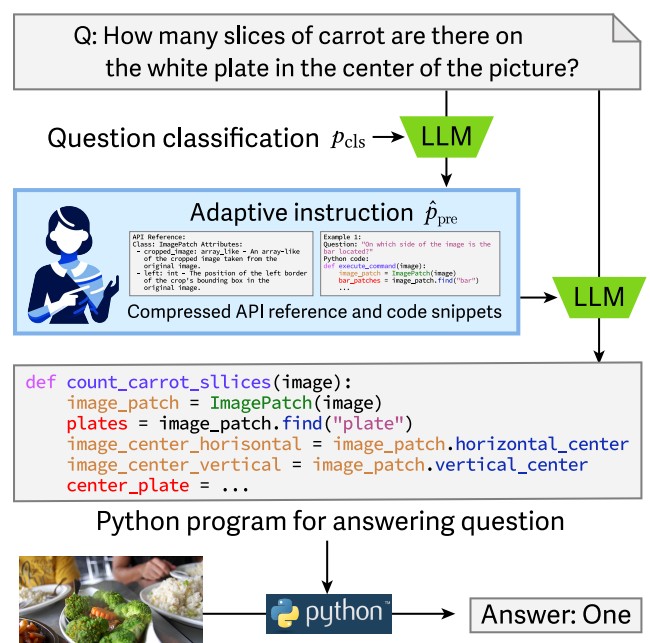

**Figure 1: Inference procedure of AdaCoder. Given an input question, an adaptive programming instruction $\hat{p}_{pre}$ for the specific question type is used to generate a Python program for visual question answering.**

and natural language processing [2, 6, 11, 14, 30, 37]. Thanks to the advantages of deep learning techniques, significant progress has been made in VQA over the past decade with end-to-end learning models such as GLIP [22]. However, these models do not explicitly distinguish between visual processing and reasoning, which limits their generalizability and interpretability.

To overcome this limitation, several pioneering studies have introduced visual programmatic models (VPMs), models that generate executable programs specifically designed to answer questions, providing a more manageable and transparent inference process [12, 29, 31]. VPMs typically consist of a large language model (LLM) for code generation and a set of APIs for image processing. Given an input question, the LLM analyzes the text to understand the intent and the required computational steps. It then generates a program that, when executed, can manipulate and analyze images by using APIs to produce the desired answer, where the APIs include both low-level modules (*e.g.*, image cropping) and high-level modules (*e.g.*, object detection). VPMs have proven effective and are gaining traction; however, they also face challenges in terms of computational complexity, as long prompts are required to enable the LLM to understand API usage for generating appropriate programs.

To reduce computational costs, the development of efficient neural network architectures has been extensively studied. However, these approaches require retraining or additional learning, which is not feasible for application to LLMs trained with huge data, such as GPT and Claude, which we refer to as black-box LLMs. Recently, research has begun to focus on prompt compression [8, 15, 25], which involves optimizing input prompts to achieve high performance with shorter inputs. For example, LLMLingua [15, 26] compresses prompts using smaller models before using black-box LLMs.

Inspired by these studies, we introduce AdaCoder, a framework of adaptive prompt compression for VPMs. More specifically, Ada-Coder operates in two phases: a compression phase and an inference phase. The compression phase generates a set of compressed preprompts, each depending on a specific question type, given a preprompt that describes all API definitions in the Python language with example snippets of code. The inference phase adaptively selects a compressed prompt by classifying the question type and generates a Python program to answer the input question, as shown in Figure 1. Notably, we implement all of the modules of AdaCoder with a single frozen LLM, which allows implementation with black-box LLMs. Our contributions are as follows:

1) We propose AdaCoder, a novel prompt compression framework for VPMs. It adaptively selects a short instruction for code generation based on question type.

2) We define and formulate all procedures of AdaCoder with a single frozen LLM. This avoids additional training and enables implementation with black-box LLMs.

3) We demonstrate the effectiveness of AdaCoder over the state-of-the-art ViperGPT [31] model on three VQA datasets with GPT and Claude. We show that the token length of input prompts is reduced by 71.1%, while maintaining or even improving question answering performance. We also show that AdaCoder outperforms LLMLingua [15] in our evaluations.

## 2 RELATED WORK

### 2.1 Visual question answering

**End-to-end models.** In the early phase of VQA research history, a number of neural network architectures designed to process multi-modal inputs were introduced. These include a combination of a convolutional neural network (CNN) for visual feature extraction and a recurrent neural network (RNN) for textual feature extraction [13, 24, 41]. Recent models often include attention modules to enhance individual feature extraction for each modality and to combine features of multiple modalities effectively [1, 28, 36, 38]. Large-scale pre-training has become a critical component in improving the performance of these models, enabling them to answer complex questions by implicitly associating words with specific regions in images [20–22]. More recently, LLMs have been incorporated into VQA frameworks with prompt tuning techniques such as self-prompt tuning [40]. However, these models do not explicitly distinguish between visual processing and reasoning, limiting their interpretability. Some recent studies have focused on techniques to improve interpretability such as causal inference [5], reasoning path [23], reasoning prompts [19] and gradient-based explainability method [34].

**Visual programmatic models.** To improve interpretability and generalizability, VPMs that generate programs to answer questions based on visual input have been gaining research attention. This is a novel approach that leads to more manageable and traceable inferences because the generated programs contain logical sequences that are understandable to humans and articulate a step-by-step methodology for reaching conclusions. Examples of VPMs include ViperGPT [31], VisProg [12], and CodeVQA [27]. All of these generate Python programs utilizing image processing APIs, such as object detection, through a frozen LLM. However, generating programs to answer complex and compositional questions requires many APIs and example codes for them. As a result, the length of the input prompt becomes long. To the best of our knowledge, this work is the first to propose adaptive prompt compression for VPMs.

### 2.2 Large language models

**Code generation.** Extensive research and development in the field of natural language processing (NLP) has led to the creation of LLMs that excel at a variety of NLP tasks. Among these, a distinct group of LLMs is specifically designed for programming code generation, having been trained on large amounts of programs and documents related to programming. For example, Codex [7], a variant of the GPT-3 lineup, demonstrates its proficiency in multiple programming languages. CodeLlama [27], which is built on Llama2 [33] and has an expanded code dataset, shows improved performance in handling larger contexts in programming.

Most recently, black-box LLMs such as GPT-3.5/4 [4], Claude[1] and Gemini [32] integrate extensive knowledge from a broad spectrum of domains, including programming, allowing them not only to generate code but also to understand and execute complex instructions given by humans. Since their zero-shot performance on programming tasks is remarkably high, they are expected to automate many aspects of coding in future that were previously manual and time-consuming, and are also useful for integration into VPMs for visual question answering.

**Reasoning and interpretability.** Interpretability is an important consideration when integrating LLMs into real-world systems, especially in contexts that require high reliability and accountability. Various prompting techniques have significantly improved the interpretability of LLMs. For example, chain-of-thought prompting [16], which provides an LLM with a series of contextual examples, enables intermediate reasoning to reach final conclusions. Tree-of-thought prompting [39] constructs a tree structure of thoughts, enriching the decision-making process by branching out various reasoning pathways. VPMs can also be viewed as an extended prompting method that improves interpretability because they show a sequence of logical steps leading to a conclusion by understandable programs. However, these methods also increase the complexity of input prompts because the instructions for LLMs need to be detailed, thus increasing computational costs.

**Prompt compression.** Several strategies have been developed to compress prompts, notably by creating specialized tokens through prompt-based fine-tuning of LLMs [8, 9, 25, 35], with the goal of minimizing the number of tokens processed during inference. However, fine-tuning of LLMs often limits their generalizability and

---

[1]https://claude.ai

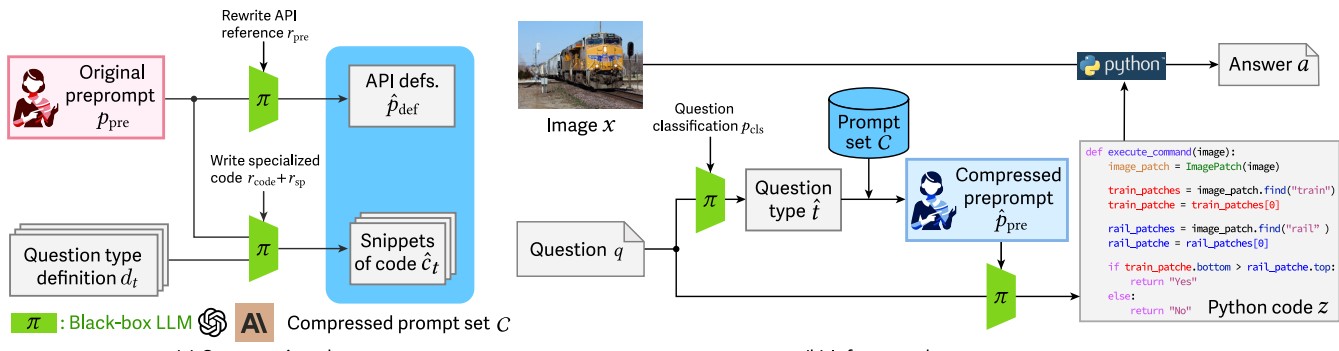

Figure 2: AdaCoder framework. (a) Compression phase generates a set of compressed prompts $C$ by utilizing an LLM $\pi$ with two instructions $r_{\text{pre}}$ for rewriting API definitions and $r_{\text{code}} + r_{\text{sp}}$ for writing snippets of code specialized for each question type $t$. (b) Inference phase adaptively selects code snippets to create compressed preprompt $\hat{p}_{\text{pre}}$ for generating a Python code $z$ for visual question answering.

is not always applicable to black-box LLMs. Other efforts have focused on token reduction. These include token pruning during inference [10, 17, 18] and token merging [3]. However, these methods are generally proposed for small models such as BERT and ViT, and rely on fine-tuning or intermediate inference results. Most recently, Jiang *et al.* [15] have introduced LLMLingua, which compresses prompts with a small model and feeds the compressed prompts to an LLM. This method can be applied to black-box LLMs because it does not require comprehensive fine-tuning of LLMs.

In contrast to these previous studies, this work aims to define and formulate all procedures of prompt compression and inference for code generation with a single frozen LLM to fully leverage the advantages of powerful black-box LLMs.

## 3 ADACODER FRAMEWORK

This section introduces AdaCoder, a framework for adaptive prompt compression for VPMs. Figure 2 shows an overview of the AdaCoder framework, which consists of two phases: the compression phase and the inference phase. The compression phase is run only once to prepare compressed prompts, each of which is specialized for a specific question type. The inference phase classifies question type and adaptively selects a compressed preprompt to generate code for visual question answering. Below, we begin with a preliminary formulation of a VPM. We then present each phase of AdaCoder.

### 3.1 Preliminary

**Notation and settings.** We follow the notation used in previous work on VPMs [29, 31]. Let $x \in X$ be an input image and $q \in Q$ be an input question about the image, where $X$ is a set of images and $Q$ is a set of questions. VPMs aim to generate a code $z \in Z$ that returns the answer $a \in A$ to the question, where $Z$ is a set of executable codes and $A$ is a set of answers.

The process of answering questions is divided into two steps: the code generation step and the execution step. The former generates a code as

$$z = \Pi(q), \tag{1}$$

where $\Pi : Q \to Z$ is a code generation module. The latter executes the code with an input image by

$$a = \Lambda(x, z), \tag{2}$$

where $\Lambda : X \times Z \to A$ is the execution engine. This work utilizes the Python execution engine for $\Lambda$.

**Large language model.** To implement the code generation module, a single frozen LLM $\pi : T \to T$ is often used, where $T$ is a set of texts[2]. For example, the code generation module $\Pi$ can be defined by

$$\Pi(q) = \pi(p_{\text{pre}} + q), \tag{3}$$

where $p_{\text{pre}} \in T$ is a preprompt that gives instructions to generate code using image and text processing APIs, $q \in Q$ is an input question, and + indicates textual concatenation. Here, APIs include both low-level functions, such as image cropping, and high-level functions, such as object detection.

**Preprompt definition.** In order to provide the LLM with detailed instructions on how to use the APIs, the preprompt $p_{\text{pre}}$ typically includes API definitions, coding instructions, and example snippets of code. We define a preprompt $p_{\text{pre}}$ by

$$p_{\text{pre}} = \Psi(p_{\text{def}}, c, p_{\text{inst}}), \tag{4}$$

where $p_{\text{def}}$ is a text of API definitions, $c \in Z$ is textually concatenated example snippets of Python code, $p_{\text{inst}} \in T$ is a coding instruction written in a natural language, and $\Psi$ is a structural aggregation function to insert code snippets to immediately after function definitions as comments. For example, a code snippet for comparing the positions of objects is inserted immediately after the definition of the object detection function. Below, we review the preprompt of ViperGPT [31], which we use in Section 4.

*1) API definitions.* The text of API definitions for $p_{\text{def}} \in Z$ is written in Python and includes both class, method and function definitions. Specifically, it consists of the Python class `ImagePatch` to represent an image patch and a set of auxiliary functions.

*2) Code snippets.* For each function and method, one or two code snippets are provided. Each code snippet calls the function or

---

[2]This work assumes that questions, answers, and codes are in text form, *i.e.*, $Q$, $A$, and $Z$ are subsets of $T$.

method at least once. For example, the following code snippet is given for the `find` method that detects objects in images.

```
# Return the foo
def execute_command(image) -> List[ImagePatch]:
    image_patch = ImagePatch(image)
    foo_patches = image_patch.find("foo")
    return foo_patches
```

*3) Coding instruction.* The coding instruction provides short instructions describing how to write code, specifying a programming language and how APIs should be used. Specifically, $p_{inst}$ is the following text:

```
Write a function using Python and the ImagePatch class (above) that
could be executed to provide an answer to the query.

Consider the following guidelines:
- Use base Python (comparison, sorting) for basic logical
  operations, left/right/up/down, math, etc.
- Use the llm_query function to access external information and
  answer informational questions not concerning the image.
```

**Token length.** One of major limitations of previous VPMs is that the input token length is long, resulting in a large computational load. This work addresses this limitation by introducing an adaptive prompt compression method. More specifically, we define the input token length of the code generation module in Eq. (3) as

$$\ell(q; \Pi) = |p_{pre}| + |q|, \tag{5}$$

where $|t|$ is the number of tokens of a text $t \in T$. Our goal is to reduce this length.

## 3.2 Compression phase

As shown in Figure 2a, the compression phase creates a set of compressed prompts $C = \{\hat{p}_{def}\} \cup \{\hat{c}_t : t \in Y\}$, where $\hat{p}_{def}$ is a compressed text of API definitions, $\hat{c}_t$ is a compressed code snippet for question type $t \in Y$, and $Y$ is a set of question types. Below, we detail the two-step process for compressing API definitions and code snippets.

**Compressing API definitions.** This step compresses the API definitions by

$$\hat{p}_{def} = \pi(p_{pre} + r_{pre}) \tag{6}$$

where $p_{pre}$ is the original preprompt in Eq. (4), $\pi$ is a frozen LLM, and $r_{pre}$ is the instruction to rewrite API definitions. Figure 3a shows the definition of $r_{pre}$.

**Compressing code snippets.** This step compresses the code snippets for each question type as follows:

$$\hat{c}_t = \pi(p_{pre} + r_{code} + r_{sp}(d_t)), \tag{7}$$

where $p_{pre}$ is the original preprompt, $r_{code}$ is the instruction to write code snippets, $r_{sp}$ is an additional instruction to write code specialized for a specific question type with a placeholder to insert the definition of question type $d_t$, and $t \in Y$ is a question type. Figure 3b and 3c show the definitions of $r_{code}$ and $r_{sp}$, respectively. Here, we assumed that a pre-defined set of question types $Y$ is given. For example, with the GQA dataset [14], five question types shown in Table 1 are provided with their definitions.

**(a) Instruction to rewrite API definitions $r_{pre}$**

```
Rewrite the API reference above.

Consider the following guidelines:
- Make the API reference shortly.
- Must include all class, methods, fields, and functions above.
- Must include some explanations to make clear the usage of each
  class, methods, fields, and functions above.
```

**(b) Instruction to write code snippets $r_{code}$**

```
Write some examples of a question and Python code using the API
reference above that provides the answer to the question.

Consider the following guidelines:
- Must use all class, methods, fields, and functions above at least
  one time.
- Must refer to code example above.
- Must write at least three example.
```

**(c) Instruction to specialize for specific question type $r_{sp}$**

```
- Must make {type_definition[i]}
```

**Figure 3: Instruction prompts for the compression phase.** `{type_definition[i]}` **is a placeholder to which a question type definition $d_t$. See Table 1 for example type definitions.**

## 3.3 Inference phase

The inference phase generates a code to answer the input question by utilizing a compressed preprompt adaptively selected based on the question type as shown in Figure 2b. More specifically, this phase consists of four steps: question classification, preprompt generation, code generation, and execution.

**Question classification.** Given an input question $q \in Q$, this step predicts the question type. We define classification prompt $p_{cls}$ and use the LLM $\pi$ for question classification as follows:

$$\hat{t} = \pi(p_{cls} + q), \tag{8}$$

where $\hat{t}$ is the predicted question type. The classification prompt consists of a short instruction for classification and a list of definitions of question types. The definition of classification prompt is shown in Figure 4.

**Preprompt generation.** This step generates a compressed preprompt given the question type as follows:

$$\hat{p}_{pre} = \Psi(\hat{p}_{def}, p_{inst}, \hat{c}_{\hat{t}}) \tag{9}$$

where $\hat{p}_{def} \in C$ is the compressed API definitions in Eq. 6, $p_{inst}$ is the coding instruction, $\hat{c}_{\hat{t}} \in C$ is the snippets of code for the question type $\hat{t}$, and $\Psi$ is the structural aggregation function in Eq. (4). Note that the computational cost of this step is almost negligible because both compressed prompts, $\hat{p}_{def}$ and $\hat{c}_{\hat{t}}$, have already been computed in the compression phase. We do not compress the coding instruction $p_{inst}$, because it is already short.

**Code generation.** This step generates a Python code $z$ as follows:

$$z = \pi(\hat{p}_{pre} + q), \tag{10}$$

where $\hat{p}_{pre}$ is the compressed preprompt.

**Execution.** Finally, the predicted answer $a$ to the question is obtained by executing the code as follows:

$$a = \Lambda(x, z), \tag{11}$$

where $x$ is an input image, and $\Lambda$ is the Python execution engine.

**Table 1: Question type definition for the GQA dataset.**

| Type $t$ | Definition $d_t$ |
|---|---|
| obj | question asking existence of object. |
| cat | question related to object identification within some category. |
| attr | question asking about the attributes or position of an object. (e.g. "What is the color of bar?", "On which of image is the foo?") |
| rel | question derived from an affirmative sentence and asking about its subject or object (e.g. "What is the foo next to the baz wearing?", "Is the qux holding a quux?"). |
| global | question asking about the entire situation of the scene, such as weather or facility (e.g. "Is it foo?"). |

## 3.4 Discussion

**AdaCoder formulation.** By substituting Eqs. (8) and (9) into Eq. (10), we can finally define the code generation module $\Pi_{\text{Ada}}$ of AdaCoder as follows:

$$\Pi_{\text{Ada}}(q) = \pi \left( \Psi \left( \hat{p}_{\text{def}}, p_{\text{inst}}, \hat{c}_{\pi(p_{\text{cls}}+q)} \right) + q \right), \quad (12)$$

by which a code is generated as $z = \Pi_{\text{Ada}}(q)$. The total token length is given by

$$\ell(q; \Pi_{\text{Ada}}) = |\hat{p}_{\text{def}}| + |p_{\text{inst}}| + |p_{\text{cls}}| + |\hat{c}_{\pi(p_{\text{cls}}+q)}| + 2|q|. \quad (13)$$

Below, we discuss the computational cost and adaptiveness.

**Computational cost.** Although, in the first sight, AdaCoder seems computationally more expensive than the conventional code generation module in Eq. (3) because the LLM $\pi$ is called twice in Eq. (12); indeed, AdaCoder improves the computational efficiency in practice when a black-box LLM such as GPT or Claude is used for $\pi$ with state-of-the-art VPMs such as ViperGPT, because the input token length is significantly shortened. Compared to previous prompt compression methods such as LLMLingua, our approach is more efficient and effective because it can reduce the token length while preserving the structure of code. We will experimentally demonstrate this in Section 4.2.1.

**Adaptiveness.** A major strength of AdaCoder is that it does not require additional training to adaptively compress the preprompt. Since recent black-box LLMs exhibit quite high zero-shot performance on text processing tasks such as text classification and summarization, AdaCoder leverages these capabilities to enhance efficiency and reduce the computational costs of VPMs.

## 4 EXPERIMENTS

### 4.1 Experimental settings

**Datasets.** We use three VQA datasets for evaluation: GQA [14], VQAv2 [11], and NLVR2 [30]. The GQA dataset is designed to test a model's visual reasoning abilities, encompassing five question types: existence of objects (obj), category of objects (cat), attributes of objects (attr), relationships between subjects and/or objects (rel), and global questions (global). The VQAv2 dataset contains open-ended questions about images that require an understanding of

**Figure 4: Classification prompt for the inference phase.** {type[i]} and {type_definition[i]} **are placeholders for names and definitions of question type, respectively, for** $i = 0, 1, \cdots, n$ **where** $n$ **is the number of question types.**

visual content to generate answers. The NLVR2 dataset is designed to test a model's ability to understand complex natural language statements and their correspondence to a given pair of images. From each of these two dataset, we randomly choose 2,000 QAs[3].

**Evaluation metrics.** We use the exact match accuracy (%) for case-insensitive answers as a QA performance evaluation metric. The reduction rate (%) of the input token length is used to evaluate the compression performance.

**Baselines.** The baselines are ViperGPT [31] and LLMLingua [15] applied to it. They are state-of-the-art VPM and prompt compression method, respectively.

**Implementation details.** We implement AdaCoder on top of the official implementation of ViperGPT[4]. The API set consists of basic image and text processing functions. Specifically, it consists of the `ImagePatch` class and a set of auxiliary functions. The `ImagePatch` class is a class to store a image region and has the following nine methods.

> 1) crop, 2) overlaps_with, 3) find, 4) exists, 5) best_text_match,
> 6) verify_property, 7) simple_query, 8) llm_query, 9) compute_depth.

The auxiliary function set consists of the following four functions.

> 1) distance, 2) best_image_match, 3) bool_to_yesno, 4) coerce_to_numeric.

Each method or function is provided with its definition in Python and example code snippets. See the Appendix for more details.

**LLMs.** We use GPT and Claude for both ViperGPT and AdaCoder. For GPT, we use `gpt-3.5-turbo`, released as version 1106, which is trained on data up to September 2021 and is provided by the OpenAI platform. For Claude, we use `claude-3-haiku`, released as version 20240307. This is a model trained using large amounts of feedback on long document tasks. Note that the original ViperGPT used `code-davinci-002` (the GPT-3 Codex [7]), which was fine-tuned for code generation tasks and is no longer accessible.

## 4.2 Experimental results

### 4.2.1 Main results

**QA accuracy.** Table 2 shows QA accuracy in comparison to the ViperGPT baseline. We see that AdaCoder reduces the input token length by 71.1%, while improving QA accuracy on all of the three

---

[3]This is due to the usage limits of Claude and GPT. The list of sampled QA IDs will be provided along with our code.
[4]https://github.com/cvlab-columbia/viper

**Table 2: Comparison with other methods. AdaCoder is compared with ViperGPT [31], LLMLingua [15], and Simple compression that omits QA classification prompts.**

| Method | LLM | Accuracy (%) | | | Input prompt | | | Output |
| --- | --- | --- | --- | --- | --- | --- | --- | --- |
| | | GQA | VQAv2 | NLVR2 | Token length ↓ | Characters ↓ | Reduction ↑ | Token length |
| ViperGPT baseline | gpt-3.5-turbo | 41.3 | 42.7 | 59.2 | 3,434 | 15,950 | - | 78 |
| LLMLingua | gpt-3.5-turbo | 39.1 | 45.2 | 47.3 | 2,536 | 11,507 | 26.2% | 71 |
| Simple compression | gpt-3.5-turbo | 28.9 | 42.6 | 50.3 | 810 | 3,553 | 76.4% | 80 |
| AdaCoder (Ours) | gpt-3.5-turbo | **43.6** | **46.2** | **60.8** | 993 | 4,343 | 71.1% | 77 |
| ViperGPT baseline | claude-3-haiku | 40.4 | 42.6 | **60.1** | 3,777 | 15,950 | - | 300 |
| LLMLingua | claude-3-haiku | 37.0 | 43.1 | 59.5 | 2,766 | 11,507 | 26.8% | 306 |
| Simple compression | claude-3-haiku | 14.5 | 23.6 | 54,3 | 1,181 | 4,535 | 68.7% | 245 |
| AdaCoder (Ours) | claude-3-haiku | **41.6** | **44.7** | **60.1** | 1,170 | 4,503 | 69.0% | 234 |

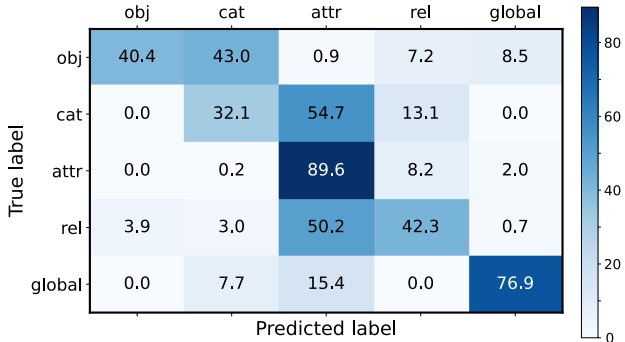

**Figure 5: Confusion matrix of question classification. Overall classification accuracy was 58.1%. (GQA dataset, gpt-3.5-turbo)**

datasets. This shows the effectiveness and efficiency of the proposed prompt compression method.

The simple compression setup omits the instruction prompt $r_{sp}$ to compress for specific question type (*i.e.*, question type classification is omitted). We see that the QA accuracy is significant degraded by this omission, which confirms the effectiveness of our adaptive approach.

With LLMLingua, we observed that it cannot maintain the structure of code snippets in the preprompt after compression at a reduction rate of 71.1% (the same rate as ours), resulting in a QA accuracy of 0%. Therefore, the results in Table 2 are given at a lower reduction rate $\simeq$ 25% by adjusting the compression ratio parameter accordingly. With this setting, LLMs can generate executable code with a probability of 98%; however, the QA accuracy is degraded by 2.2 points on GQA. This shows that prompt compression for VPMs is challenging, and that our approach specialized for preserving code structure is effective.

**Question type classification.** Figure 5 shows the confusion matrix of question type classification for the GQA dataset. We observe that two question types, "attr" and "global", achieve accuracies greater than 75%. The types "rel" and "obj" are often misclassified as "attr" and "cat", respectively. This is because the questions are often short, making it difficult to distinguish between them.

To investigate how these classification errors affect the final QA accuracy, Table 3 compares AdaCoder using 1) predicted question

**Table 3: Analysis on effect of question type classification.**

| Method | Token length | Accuracy (%) |
| --- | --- | --- |
| w/ Predicted question types | 992 | 43.6 |
| w/ Ground-truth question types | 851 | 44.5 |
| w/ Random question types | 851 | 37.6 |
| w/o Q. type based compression | 732 | 28.9 |

types, 2) ground-truth question types, 3) random question types, and 4) without using question type based compression. We observe three key findings. First, the best performance is achieved by using ground truth question types. This highlights the importance of classifying question types to improve overall accuracy. Second, the performance drop due to classification errors is less than 1.0 points. This suggests that AdaCoder effectively classified the critical question types necessary for code generation, even though the accuracy for question classification is not very high. Third, the method using random question types, which compresses prompts for each question type and randomly choose one of them in inference, is better than the method without question type based compression. This is because the instruction prompt $r_{sp}$ in Eq. (7) for specializing code snippets to each question type makes it more likely to provide code snippets that are related to each other, thereby increasing the probability of completing the program. When this instruction is omitted and compression is performed regardless of the question type, code snippets that are effective for any question type tend to be retained after compression. However, this approach results in the loss of some specific snippets that are necessary to complete the program, thereby reducing QA accuracy. These results suggest that the instruction $r_{sp}$ is important for compressing code snippets.

**Compressed prompts.** Table 4 summarizes the token length and compression performance for each component of the input preprompt. We see that both API definitions and code snippets are significantly compressed. A comparison of the original and compressed API definitions is shown in Figure 6. We see that descriptions of methods unnecessary for coding, such as those for the initialization method, are omitted, and the remaining sections are condensed into shorter sentences. This is an effective compression achieved by the language understanding and summarization capabilities of black-box LLMs.

**Computational time.** Since the model weights and details of the black-box LLMs are not publicly available, and API response times

**Table 4: Token length and number of characters for each component of input prompt. Reduction rate is measured by token length.**

| Component | ViperGPT | | AdaCoder | | |
|---|---|---|---|---|---|
| | Tokens | Characters | Tokens | Characters | Reduction |
| API defs | 1,971 | 9,299 | 541 | 2,360 | 72.5% |
| Code snippets | 1,386 | 6,263 | 233 | 971 | 76.0% |
| Instruction | 77 | 388 | 77 | 388 | - |
| Classification | 0 | 0 | 141 | 618 | - |
| Total | 3,434 | 15,950 | 992 | 4,337 | 71.7% |

```python
import math

class ImagePatch:
    """"A Python class containing a crop of an image centered around a
    particular object, as well as relevant information.
    ...

    def __init__(self, image, left: int = None, lower: int = None,
                 right: int = None, upper: int = None):
        """"Initializes an ImagePatch object by cropping the image at the given
        coordinates and stores the coordinates as attributes. If no
        coordinates are provided, the image is left unmodified, and the
        coordinates are set to the dimensions of the image.
        ...

    def best_image_match(list_patches: List[ImagePatch], content: List[str],
                         return_index = False) -> Union[ImagePatch, int]:
        """"Returns the patch most likely to contain the content.
        Parameters
        ----------
        list_patches : List[ImagePatch]
        ...
```
**1,971 tokens, 9,299 characters**

```
API Reference:

Class: ImagePatch

Attributes:
- cropped_image: array_like - An array-like of the cropped image taken from the
  original image.
- left: int - The position of the left border of the crop's bounding box in the
  original image.
  ...

Methods:
- find(object_name: str) -> List[ImagePatch]: Returns a list of ImagePatch
  objects matching object_name contained in the crop.
- exists(object_name: str) -> bool: Returns True if the object specified by
  object_name is found in the image, and False otherwise.
  ...

Functions:
- best_image_match(list_patches: List[ImagePatch], content: List[str],
  return_index=False) -> Union[ImagePatch, int]: Returns the patch most likely
  to contain the content.
- distance(patch_a: ImagePatch, patch_b: ImagePatch) -> float: Returns the
  distance between the edges of two ImagePatches.
  ...
```
**541 tokens, 2,360 characters**

**Figure 6: Comparison of the original and compressed API definitions ($p_{\text{def}}$ and $\hat{p}_{\text{def}}$). AdaCoder reduced the token length by 72.5%.**

can be affected by server congestion, a detailed analysis of computation times is not possible. However, the total time for experiments on the GQA dataset was reduced by 55%.

### 4.2.1 Ablation study and analysis

**Ablation study.** Table 5 presents the results of an ablation study. We see that both compression of API definitions and code snippets contribute to each other for both reducing the input token length and improving QA accuracy. Table 6 summarizes the QA accuracy obtained by using a single compressed prompt. We see that even with one prompt of either "attr" or "rel", our method achieves comparable or slightly better performance than the ViperGPT baseline (41.3%). However, using one prompt of either "obj" or "global",

**Table 5: Ablation study with respect to prompt compression (GQA dataset, gpt-3.5-turbo).**

| Method | Token length | Accuracy |
|---|---|---|
| AdaCoder | 992 | **43.6** |
| w/o compressing API defs. | 2,422 | 40.6 |
| w/o compressing code snippets. | 2,145 | 41.1 |
| w/o QA classification | 851 | 28.9 |
| w/o any compression | 3,434 | 41.3 |

**Table 6: Ablation study using a single specialized prompt during inference (GQA dataset, gpt-3.5-turbo).**

| Method | Token length | Accuracy (%) |
|---|---|---|
| AdaCoder (adaptive prompt) | 992 | **43.6** |
| w/ fixed prompt of obj | 967 | 30.9 |
| w/ fixed prompt of cat | 1,015 | 39.0 |
| w/ fixed prompt of attr | 1,008 | 41.7 |
| w/ fixed prompt of rel | 993 | 42.3 |
| w/ fixed prompt of global | 977 | 35.3 |

**Table 7: Cross question type evaluation (GQA dataset, gpt-3.5-turbo).**

| QA type | obj | cat | attr | rel | global |
|---|---|---|---|---|---|
| obj | **77.0** | 17.5 | 31.1 | 21.8 | 16.9 |
| cat | 74.5 | **45.3** | 39.9 | 28.5 | 35.4 |
| attr | 68.9 | 30.7 | **52.4** | 28.9 | **36.9** |
| rel | 70.2 | 35.8 | 50.9 | **30.3** | 33.9 |
| global | 71.1 | 31.4 | 38.0 | 24.8 | 32.3 |

the QA accuracy is significantly degraded. These results demonstrate that our adaptation approach is essential for improving QA accuracy while compressing input prompts. The detailed QA accuracy by question type is analyzed in Table 7. We see that the four compressed prompt specialized for "obj", "cat", "attr", and "rel" performed the best for corresponding questions. For the "global" questions, the prompt for "attr" was the best. This is because "global" questions are highly varied and not easily categorized. Defining fine-grained QA types would be interesting as a next step in future research.

**Error analysis.** Table 8 shows an error analysis, where we manually counted the occurrence of four types of errors. "Coding error" indicates that the generated program is not executable or returns nothing. "Cannot answer to simple query" indicates that the program is correct but the simple_query method returned a response such as "I cannot answer". "No object detected" indicates that no object is detected by the find method. "Wrong answer" indicates that the returned answer was wrong. We have two observations. First, the predominant type of error was wrong answers, and AdaCoder reduced their frequency. Second, despite AdaCoder's improvement in coding quality, there is still a 7.8% incidence of coding errors. This suggests that there is still room for improvement in instructing LLMs about API usage.

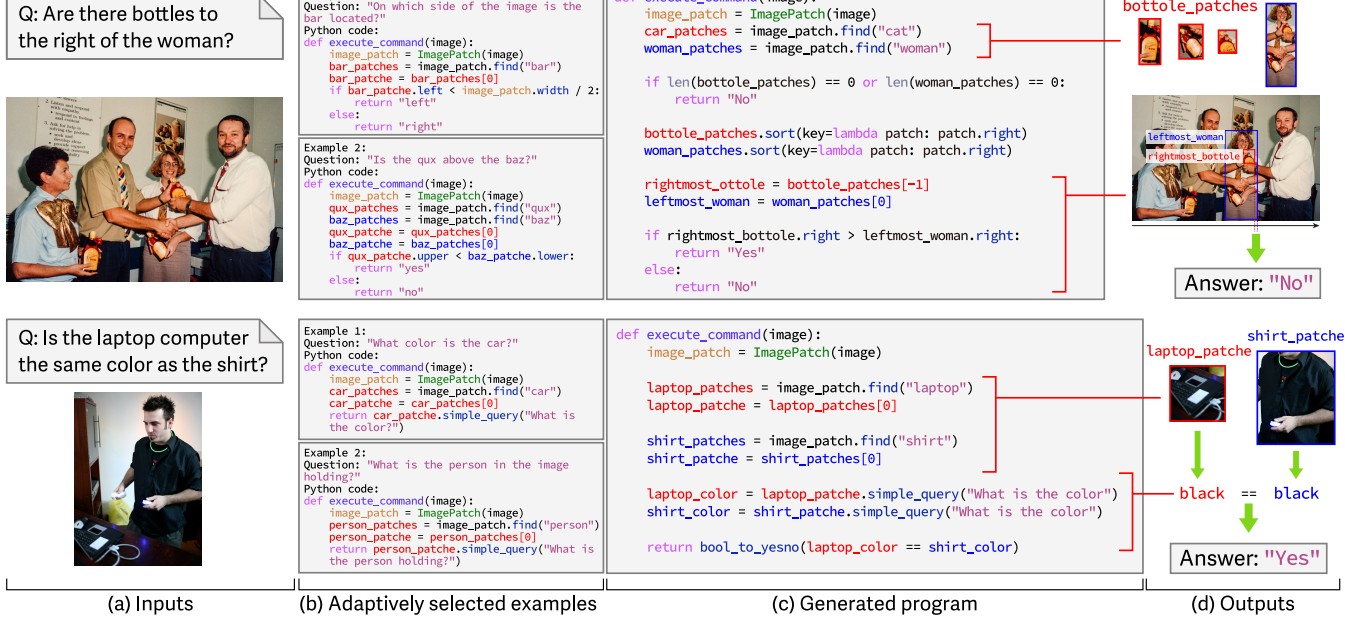

(a) Inputs  (b) Adaptively selected examples  (c) Generated program  (d) Outputs

Figure 7: Qualitative examples. (a) Input of questions and images. (b) Adaptively selected example code snippets. Each compressed prompt involves three or four snippets, and two of them are shown. These examples are fed into LLM with the compressed API definition in Figure 6. (c) Generated Python program for question answering. (d) Visualization of intermediate outputs to derive the answer.

Table 8: Error analysis (individual error rates as percentages).

| Error type | ViperGPT | AdaCoder |
| --- | --- | --- |
| Correct but with unnecessary details | 0.5 | 0.5 |
| Correct except for articles | 1.1 | 1.5 |
| Correct by paraphrasing | 1.4 | 1.6 |
| Coding error | 8.3 | 7.8 |
| Cannot answer to simple query | 6.1 | 6.1 |
| No object detected | 0.7 | 1.6 |
| Wrong answer | 40.6 | 37.3 |

Several minor errors were also observed. "Correct but with unnecessary details" refers to responses that were marked incorrect because they provided additional, unnecessary information, such as the response "Yes, there is an apple on the table" where the ground truth is "Yes". "Correct except with articles" refers to cases where the instruction to respond with a single word was ignored and an article was added, resulting in responses such as "a car" instead of "car". "Correct by paraphrasing" refers to errors resulting from the use of interchangeable terms that do not change the meaning, such as using "lady" instead of "woman".

**Qualitative examples.** Figure 7 presents qualitative examples of the generated programs. As shown, few example code snippets related to the input question are adaptively selected. These examples help LLM to generate a program to answer the question. When the program is executed, the object patches are detected and then the relative position or colors are compared to derive a correct answer.

## 5 CONCLUSION

We introduced AdaCoder, a framework for adaptive prompt compression for visual programmatic models. AdaCoder efficiently generated programs for visual question answering by compressing and selecting prompts depending on the question type. A single black-box LLM is effectively employed to perform question type classification, textual compression and code generation, eliminating the need for additional training. In experiments, we demonstrated the effectiveness and efficiency of AdaCoder in comparison to ViperGPT and LLMLingua. Finally, we discuss limitations and future work.

**Limitations.** As this work relies on black-box LLMs, analysis from the perspective of neural network architecture is limited. Alternative choices to LLMs for code generation may include open-source white-box models, such as CodeLlama and StarCoder. However, since AdaCoder requires high-quality text classification and summarization, these models were not suitable for prompt compression. New research directions leveraging the combination of white-box and black-box LLMs need to be further explored.

**Future work.** To advance multimodal automated programming, future research directions that focus on pushing the boundaries beyond the traditional scope of VQA would be interesting. This includes developing methods for interactive code modification to enable a more dynamic and responsive programming environment. Additionally, we plan to explore the automatic extension of APIs to facilitate their evolution in becoming more efficient and effective in addressing the complex requirements of multimodal interactions. We believe that the present work contributes to fostering new ideas for such novel research directions for the multimedia community.

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
