# OpenReview forum: "AdaCoder: Adaptive Prompt Compression for Programmatic Visual Question Answering"
_acmmm.org/ACMMM/2024/Conference — MM2024 Poster_

### Official Review · Reviewer_MnMj · 2024-05-23

**Rating:** 2
**Confidence:** 3

**Summary:**

This work addresses the issue of excessively long prompts in Visual Programmatic Models (VPMs) caused by APIs and example code. It proposes a solution in the form of AdaCoder, a novel prompt compression framework designed specifically for VPMs.

**Strengths:**

1. AdaCoder adaptively selects concise instructions for code generation based on the type of query.
2. This approach eliminates the need for additional training and allows for implementation with black-box large language models (LLMs).

**Limitations:**

1. The novelty of this methodology is limited, as it merely involves compressing and invoking predefined prompt sets using existing models.
2. My major concern about this paper is that focusing this VPM method solely on Visual Question Answering (VQA) tasks limits its contribution significantly. Previous methods, such as Visprog [1], ViperGPT [2], and GENOME [3], have demonstrated their improvements across various multimodal tasks, as the strength of VPM methods lies in their generality and ability to iterate and update [3]. Whether the compression method proposed in the paper can be applied to different multimodal tasks is worth further consideration.
3. In Section 3.2, Do the compressed prompt words have a dependency on the model? It is unclear whether compressing prompts with different models impacts inference. For instance, does compressing with Claude and inferring with GPT affect the experimental results?

[1] Gupta T, Kembhavi A. Visual programming: Compositional visual reasoning without training[C]//Proceedings of the IEEE/CVF Conference on Computer Vision and Pattern Recognition. 2023: 14953-14962.

[2] Surís D, Menon S, Vondrick C. Vipergpt: Visual inference via python execution for reasoning[C]//Proceedings of the IEEE/CVF International Conference on Computer Vision. 2023: 11888-11898.

[3] Chen Z, Sun R, Liu W, et al. GENOME: GenerativE Neuro-symbOlic visual reasoning by growing and reusing ModulEs[J]. arXiv preprint arXiv:2311.04901, 2023.

**Suitability:**

2

---

### Official Review · Reviewer_FB2n · 2024-05-24

**Rating:** 3
**Confidence:** 4

**Summary:**

This paper proposes AdaCoder, an adaptive prompt compression framework for visual programmatic models. AdaCoder operates in two phases: a compression phase that generates compressed prompts for different question types, and an inference phase that predicts the question type and uses the corresponding compressed prompt to generate code to answer the question.  By employing compressed prompts, AdaCoder reduces token length while maintaining or improving performance.  Experiments on ViperGPT demonstrate that AdaCoder highly reduces token length while maintaining the performance.

**Strengths:**

1.	It works well in reducing token length and can be adapted to multiple LLMs.
2.	The motivation is straightforward and clear.

**Limitations:**

1.	The challenge of long prompts tokens is not clear. Why does this need to be emphasized, and does it cause issues beyond computational complexity?
2.	In the experiments, we found that direct token compression (w/o considering Q types) still results in significant performance degradation. This is not consistent with the authors' claimed conclusion, not that compressing the token still improves performance, but that the prompts to the Q types mitigate the performance degradation.
3.	The authors need to add this experimental setup: add the prompts to the Q types on top of the other existing methods to prove the effect of token compression.
4.	The experiment found considerable improvement in adding prompts for question types, but the focus of the paper is not on the question types. Besides, it seems that the authors' design did not sufficiently distinguish the correct question types (Fig. 5).

**Suitability:**

3

---

### Official Review · Reviewer_f2r6 · 2024-05-25

**Rating:** 5
**Confidence:** 3

**Summary:**

The paper titled "AdaCoder: Adaptive Prompt Compression for Programmatic Visual Question Answering" introduces a novel approach to enhancing visual programmatic models (VPMs) by employing an adaptive prompt compression framework. This method, named AdaCoder, operates in two phases: a compression phase and an inference phase. During the compression phase, the system generates compressed preprompts tailored to specific question types. In the inference phase, AdaCoder selects the appropriate preprompt based on the input question, enabling efficient code generation for visual question answering.

Key contributions of AdaCoder include a significant reduction in token length—by 71.1%—without compromising performance. The framework uses a single frozen large language model (LLM) to handle all phases, thus avoiding the need for additional training and maintaining adaptability across different LLMs. Experimental results demonstrate that AdaCoder outperforms existing methods like ViperGPT and LLMLingua in terms of both efficiency and accuracy.

Overall, AdaCoder presents a robust solution for reducing computational load and improving the performance of VPMs, making it a valuable contribution to the fields of computer vision and natural language processing.

**Strengths:**

One of the strengths of this paper is its unique motivation. In the context of the vast number of papers on in-context learning for VQA using large language models (LLMs), this paper stands out with its innovative approach. The authors identify the significant challenge of lengthy input prompts required for LLMs in visual programmatic models and address it with a novel adaptive prompt compression framework. This fresh perspective not only contributes to the existing body of knowledge but also opens new avenues for research in optimizing prompt efficiency and effectiveness in VQA tasks.

Additionally, the experimental validation is thorough and robust. The authors have conducted extensive experiments across multiple datasets, demonstrating that AdaCoder effectively reduces token length by 71.1% while maintaining or even improving performance. The comparative analysis with state-of-the-art methods like ViperGPT and LLMLingua provides compelling evidence of the framework's superiority in terms of both efficiency and accuracy. This comprehensive experimental evaluation reinforces the practical applicability and potential impact of AdaCoder in the field of visual question answering.

**Limitations:**

One limitation of the paper, ironically, also lies in its unique motivation. While the approach of reducing input token length through adaptive prompt compression is innovative and well-executed, its long-term significance is uncertain. As large language models continue to evolve, there is a clear trend toward supporting larger context windows. This raises the question of whether efforts to shorten input prompts will remain relevant. In the future, as LLMs are likely to handle increasingly larger context windows, the necessity and impact of reducing token length may diminish. This potential shift in the development trajectory of LLMs casts some doubt on the lasting value of the proposed solution.

**Suitability:**

3

---

### Meta-Review · Area_Chair_HsB6 · 2024-07-01

**Recommendation:** Accept (Poster)
**Confidence:** 2

**Metareview:**

The paper presents AdaCoder, an adaptive prompt compression framework for visual programmatic models, which adaptively selects concise instructions for code generation based on the type of query. AdaCoder effectively reduces token length during inference on the experimented VQA tasks.

Reasons to accept:
The motivation of prompt compression in LLMs and multimodal tasks is valid and clear. The paper is a pioneer work in exploring the compression perspective in Programmatic VQA.

Major concerns:
1. The long-term significance of prompt compression remains unclear. (Reviewer f2r6, FB2n - Q1)

AC agrees that the long-term importance of prompt compression remains unclear given the increasing LLM context window. However, AC thinks we should not penalize novel explorations due to such uncertainty, the discussions in rebuttal L15 provides a valid explanation on why such technique could be useful.

2. The influence of question type (question classification accuracy) on the final performance (Reviewer FB2n - Q2)

AC agrees with the concerns raised in Reviewer FB2n’s final rating justification, that the current method heavily relies on the QA classification, instead of already being a robust compression approach across all problems.
Instead of dodging the problem or overclaiming the effectiveness, authors should comprehensively discuss and analyze such limitations to inspire future works in closing the gap. This may include the oracle performance with best QA results on different tasks to understand the cause of failures, and how different types of problems are affected by the categories.

3. The effectiveness on general VL tasks, beyond VQA (Reviewer MnMj - Q2).

AC thanks authors for adding Table 1 in rebuttal for results on visual reasoning, image editing, visual grounding, counting, relative depth estimation, and object tagging. Authors are encouraged to organize the results into the revision, with the analysis similar to the ones in the main paper and in earlier questions.

The paper receives the initial rating of weak reject, borderline reject, and weak accept. After the rebuttal, reviewers decided to keep their rating. After reading the paper, reviews, and discussion, AC agrees that the paper has merit and recommends accept, **with the condition that authors address the concerns listed above**, and incorporate corresponding content into the final version.